# Multidisciplinary Neuromuscular and Endurance Interventions on Youth Basketball Players: A Systematic Review with Meta-Analysis and Meta-Regression

**DOI:** 10.3390/ijerph19159642

**Published:** 2022-08-05

**Authors:** Arnau Sacot, Víctor López-Ros, Anna Prats-Puig, Jesús Escosa, Jordi Barretina, Julio Calleja-González

**Affiliations:** 1University School of Health and Sport (EUSES), University of Girona, 17004 Girona, Spain; 2Basquet Girona, 17007 Girona, Spain; 3Research Institute of Education, Faculty of Education and Psychology, University of Girona, 17004 Girona, Spain; 4Research Group of Clinical Anatomy, Embryology and Neuroscience (NEOMA), Department of Medical Sciences, University of Girona, 17004 Girona, Spain; 5Girona Biomedical Research Institute (IDIBGI), 17190 Salt, Spain; 6Germans Trias i Pujol Research Institute (IGTP), 08916 Badalona, Spain; 7Faculty of Education and Sport, University of Basque Country, 01007 Vitoria-Gasteiz, Spain; 8Faculty of Kinesiology, University of Zagreb, 10110 Zagreb, Croatia

**Keywords:** plyometrics, training, young, agility, basketball, sport, junior

## Abstract

The main aims of this systematic review with meta-analysis and meta-regression were to describe the effect of multidisciplinary neuromuscular and endurance interventions, including plyometric training, mixed strength and conditioning, HIIT basketball programs and repeated sprint training on youth basketball players considering age, competitive level, gender and the type of the intervention performed to explore a predictive model through a meta-regression analysis. A structured search was conducted following PRISMA guidelines and PICOS model in Medline (PubMed), Web of Science (WOS) and Cochrane databases. Groups of experiments were created according to neuromuscular power (vertical; NPV and horizontal; NPH) and endurance (E). Meta-analysis and sub-groups analysis were performed using a random effect model and pooled standardized mean differences (SMD). A random effects meta-regression was performed regressing SMD for the different sub-groups against percentage change for NPV and NPH. There was a significant positive overall effect of the multidisciplinary interventions on NPV, NPH and E. Sub-groups analysis indicate differences in the effects of the interventions on NPV and NPH considering age, gender, competitive level and the type of the intervention used. Considering the current data available, the meta-regression analysis suggests a good predictability of U-16 and plyometric training on jump performance. Besides, male and elite level youth basketball players had a good predictability on multidirectional speed and agility performance.

## 1. Introduction

Basketball is a court-based team sport that involves high technical, tactical and physical demands performed in a very short period of time with limited time-frames for recovery [1,2,3]. From a conditioning standpoint, basketball is an intermittent sport involving aerobic and anaerobic capacity needs with numerous explosive accelerations, decelerations, jumps, multidirectional sprints and change of directions [4,5,6,7,8]. Therefore, high levels of strength and neuromuscular power together with endurance levels to repeat and recover from this high intensity actions will be required for success in this sport at a senior and junior level [1,3,9,10,11,12,13,14,15]. High participation is registered in basketball across all age ranges, involving numerous benefits for health such us bone mineral density, body mass regulation and metabolic systems development, especially in youth [16,17,18]. However, high levels of practice are associated with high injury risk and in fact, the overall injury rate in youth athletes is 14.4 injuries per 1000 h of practice [16,19,20]. For this reason, considering all the aforementioned, the implementation of accurate training methods to maximize the development of physical demands and avoid injury risk are of significant interest to basketball coaches for youth player’s development [16].

Traditionally, short explosive-type training has been implemented to improve power and speed among athletes in different sports. In particular, plyometric training with different types of jumping actions is able to improve power and speed in different sports [21,22,23,24,25]. Similarly, sprint training is shown to also enhance power, speed and jump performance [24,26]. In basketball, all those explosive actions need to be performed in different movement vectors to excel by the different actions demanded during competitive conditions [1,5]. Therefore, all movement vectors should be emphasized in training periodization in order to integrally develop basketball players [3,9,27,28,29]. Additionally, continuous and intermittent endurance training along with high intensity interval training (HIIT) and repeated sprint ability (RSA) are claimed to be the best methods in order to improve aerobic and anaerobic fitness in basketball [9,10,30,31]. However, special adaptations should be considered in youth athletes and specially according to their development stage and sex [32,33,34,35,36]. There are critical and sensitive periods for physical capacity development linked to physiological changes due to maturational development with an ongoing chronological age [32,33,35,36,37,38].

Literature with respect to youth athletes, considering age, gender and competitive level of the participants provide clear information about the best training interventions [39] in order to improve endurance [40], jump performance [41] and multidirectional speed and agility [42] in team sports. Nonetheless, from the best of the author’s knowledge, there is no systematic review with meta-analysis and meta-regression jointly analyzing the effectiveness of different multidisciplinary interventions on youth basketball players.

For this reason, the main aim of this systematic review with meta-analysis and meta-regression were (1) to describe the effect of multidisciplinary neuromuscular and endurance interventions on performance in jump, multidirectional speed and endurance in youth basketball players (2) to analyze the magnitude of the effect considering age, competitive level, gender of the basketball players and according to the type of the intervention performed, and (3) to explore a predictive model to explain these effects considering age, gender, competitive level and training type through a meta-regression analysis.

## 2. Materials and Methods

### 2.1. Literature Search Strategies

This systematic review was carried out following the guidelines of the Preferred Reporting Items for Systematic Review and Meta-Analyses (PRISMA^®^) [43] and the PICOS model for definition of the inclusion criteria [44]. The PICOS model is shown in Table 1.

A structured search was conducted in Medline (PubMed), Web of Science (WOS) and Cochrane databases with no date restriction up to March 2021. Keywords were collected through expert opinion, literature review, and controlled vocabulary (e.g., Medical Subject Headings [MeSH]). Free-text words for key concepts associated with both youth and basketball were used resulting in the following unique search equation: ((“basketball” [MeSH Terms]) AND ((“young” [All Fields]) OR (“youth” [All Fields]))). Through this equation, all relevant articles in the field were obtained. The reference sections of all identified articles were also examined by applying the *“snowball methods”* strategy, based on examining the reference sections of the identified articles [45]. No other terms were used to explore the whole literature related to the topic and increase the power of the analysis, collecting all available data to date.

All titles and abstracts from the search were cross-referenced to identify duplicates and any potential missing studies (A.S. and J.E.). Titles and abstracts were selected for further review of the full text. The search for published studies was carried out independently by 2 authors (A.S. and J.E.) and in case of disagreements were resolved through discussion with third party group’s experts (J.C.-G. and V.L.-R.).

### 2.2. Study Selection

One reviewer (A.S.) searched the databases and selected all studies. Four more reviewers (V.L.-R., A.P.-P., J.C.-G. and J.B.) were available to help with study eligibility. No disagreements about the appropriateness of an article were encountered.

### 2.3. Inclusion and Exclusion Criteria

For the articles obtained in the search, the following inclusion criteria were applied to final selected studies: (1) Studies published in peer-reviewed journals and full text available (2) controlled and uncontrolled interventions with a two group design involving more than 4 week intervention, (3) carried out only in youth basketball players from 10 to 18 years old, (4) performing on field neuromuscular and endurance interventions related to strength, power, endurance or speed/agility, (5) including neuromuscular and endurance assessments as a dependent variable, (6) performed on any number or type of basketball player regardless of category, experience, competitive level or sex and (7) published in Spanish or English. The following exclusion criteria were applied to the experimental protocols of the investigation: (1) studies involving senior basketball players 18 years old, (2) interventions performed in lab conditions; (3) the absence of reliable measurements and (4) studies conducted in participants with a pathological condition.

### 2.4. Data Extraction

Once the inclusion/exclusion criteria were applied to each study, experiments were clustered by the type of test used to assess the neuromuscular and endurance changes produced by intervention. Group of experiments were created according to endurance and neuromuscular power in different vectors (vertical and horizontal). Studies involving jump assessments as a dependent variable were considered as neuromuscular power vertical (NPV), whereas studies employing sprint and agility assessments as dependent variables were considered as neuromuscular power horizontal (NPH). All endurance (E) studies included Yo-yo intermittent recovery test as assessment for the dependent variable. Data on study source includes: authors, year of publication, study design, sample size, characteristics of the participants (competitive level, age and gender), intervention (type and duration), assessment test and final outcomes of the interventions. All were extracted independently by the main author (A.S.) using a spreadsheet (Microsoft Inc., Seattle, WA, USA).

Mean (M) and standard deviation (SD) data from pre and post measurements were extracted from the tables of all the included articles to compute effect sizes. Data and sub-groups formation were discussed with the research group until consensus (V.L.-R., A.P.-P., J.C.-G., J.E. and J.B.).

### 2.5. Risk of Bias Assessment

Methodological quality and risk of bias were assessed independently by 2 authors (A.S., J.E.) following Cochrane Collaboration Guidelines [46]. In case of disagreements, they were resolved by third-party evaluation (V.L.-R., A.P.-P., J.C.-G., J.B. and J.E.).

In the Cochrane Collaboration Guidelines, the items on the list were divided into seven different domains: random sequence generation (section bias), allocation concealment (section bias), blinding of participants and personnel (performance bias), blinding of outcome assessment (detection bias), incomplete outcome data (attrition bias), selective reporting (reporting bias), and other types of bias. Then, we characterized as “low”, if criteria for a low risk of bias were met (plausible bias unlikely to seriously alter the results) or “high” if criteria for a high risk of bias were met (plausible bias that seriously weakens confidence in the results). If the risk of bias was unknown, it was considered “unclear” (plausible bias that raises some doubts about the results). Figure 1, illustrate the summary of the risk of bias assessment for the included studies for qualitative analysis.

### 2.6. Sub-Groups Analyses

Sub-groups were made according to competitive level, chronological age and gender of the basketball players and also considering the type of the intervention for both NPV and NPH. No sub-groups analysis were created in E due to the lack of studies.

Competitive level of the participants was considered elite when the basketball players were described as elite level or participated in national level competition. Conversely, basketball players from schools or regional levels were considered as amateur [47]. The chronological age categories were established according to the data from the included studies. The U-12 age category included all the participants with an age between 10 and 12 years, U-14 category included participants between 13 and 14 years, U-16 category included participants between 15 and 16 years and U-18 age group included participants between 17 and 18 years [48]. In case of studies included both (U-16 and U-18 together) without discriminating data within the study, were analyzed as U-18.

Interventions were divided among mixed strength and conditioning training, plyometric training and endurance. Mixed strength and conditioning sub-groups included all the interventions described as mixed strength and conditioning [49] and also the interventions involving bilateral and unilateral strength [50], lower body repeated power training [51] and hang clean or half squat training [52]. All the interventions involving jump protocols were included in plyometric training sub-groups. All the endurance-related protocols were included in the endurance sub-groups.

### 2.7. Meta-Regression

In order to explore a prediction model to explain the percentage change in the different interventions according to age, gender, competitive level and training type, a random effects meta-regression was performed. The SMDs from the different sub-groups were individually regressed against the percentage change between pre and post in each study, considering SMDs as an independent variable and percentage change as a dependent variable in the analysis. All the sub-groups with 4 or more studies were meta-regressed to provide a complete picture of the model in each group. In order to reduce the risk of identifying false associations in the meta-regression analysis, only the models including a minimum of 10 studies were considered in the results [46,53].

Meta-regression analyses were conducted in Statistical Package for Social Science (SPSS; V.25.0, Chicago, IL, USA) using Wilson’s SPSS macros to build all regression models [54,55]. Adjusted R^2^ was considered in the moment to quantify the proportion of variance in the model that is predicted by the independent variable. A R^2^ between 0 and 0.4 was considered as high accountable variance, between 0.4 and 0.6 moderate, between 0.6 and 0.8 small and above 0.8 very small accountable variance in the model [56]. Additionally, sum of squares indicated the significance of the model predicting changes in percentage gain according to the different sub groups [56].

### 2.8. Statistical Analyses

Descriptive data of the participants are reported as the mean ± SD. Meta-analytic statistics and figures of risk of bias were created with Review Manager© (RevMan) version 5.3 (The Nordic Cochrane Centre, The Cochrane Collaboration: Copenhagen, Denmark, 2014).

Mean, SD and sample size of experimental group of each study were used to quantify changes in pre to post measures in sub-groups. Pre-measurements on each experimental group for each study were used as a control. Standardized mean differences (SMD) for each study group were calculated using Hedges’s g [57], in which mean differences were weighted by the inverse of variance to calculate an overall effect and its 95% confidence interval (CI). It was decided to use a random effects model with the DerSimonian and Laird method [58] in order to reduce heterogeneity issues. Cohen’s criteria were used to interpret the magnitude of SMD: <0.2, trivial; 0.2–0.5, small; 0.5–0.8, moderate; and >0.8, large [59].

To avoid problems using Q statistic to assess systematic differences (heterogeneity), it calculated the I^2^ statistic, which indicated the percentage of observed total variation across studies that was due to real heterogeneity rather than chance [60]. An I^2^ value between 25% and 50% represents a small amount of inconsistency, an I^2^ value between 50% and 75% represents a medium amount of heterogeneity, and an I^2^ value > 75% represents a large amount of heterogeneity [60].

## 3. Results

### 3.1. Main Search

A total of 22 studies [49,50,51,52,61,62,63,64,65,66,67,68,69,70,71,72,73,74,75,76,77,78] were identified for inclusion in this systematic review and meta-analysis. From the final selection, 15 studies were included in NPV [49,51,52,62,63,66,68,70,71,72,73,74,75,76,78], 13 in NPH [49,50,51,61,62,66,67,69,70,71,74,77,78] and 3 in E [64,65,79]. Table 2 shows the data summary from the included studies. Figure 2 illustrates the flow diagram for study selection.

### 3.2. Risk of Bias Assessment

Results for risk of bias assessment are illustrated in Figure 1. Low risk of bias was evident in the analysis following the Cochrane Collaboration Guidelines assessment. The primary source of bias detected was found in selection bias and other bias domains.

### 3.3. Neuromuscular Interventions in Youth Basketball Players

All the included studies were randomized controlled designs with pre and post measurements investigating the effect of multidisciplinary neuromuscular interventions on endurance and neuromuscular power (vertical and horizontal) in youth basketball players. A total of 528 participants including 399 males and 129 females participated in the selected studies. The description of the participants and interventions is described in Table 3.

### 3.4. Effect on Neuromuscular Power Vertical

Figure 3 illustrates the forest plot for NPV. Thirty trials [49,50,52,62,63,64,66,67,68,71,72,73,74,75,76,77,78] involving 320 participants were included for the analysis. There was a significant small to moderate positive effect of the multidisciplinary interventions on NPV with no heterogeneity (SMD, 0.47; 95% CI: 0.31 to 0.63; I^2^, 0%; Z, 5.74; *p* < 0.0001).

#### 3.4.1. Age

Figure 4A shows the forest plot for the age sub-groups analysis on NPV. Low level of heterogeneity was found in all the analysis except from the U-18 group with moderate level (I^2^, 43%; *p* = 0.08). No significant effect was found in the U-12 sub-groups (SMD, 0.51; 95% CI: −0.28 to 1.29; I^2^, 0%; Z, 1.26; *p* = 0.21) and also in the U-14 sub-groups (SMD, 0.34; 95% CI: −0.02 to 0.70; I^2^, 1%; Z, 1.85; *p* = 0.06). A significant positive moderate effect was found in the U-16 group (SMD, 0.56; 95% CI: 0.33 to 0.80; I^2^, 0%; Z, 4.71; *p* < 0.0001). Significant positive small effect was found in the U-18 group (SMD, 0.40; 95% CI: 0.11 to 0.69; I^2^, 43%; Z, 2.67; *p* = 0.008).

#### 3.4.2. Gender

Figure 4B displays the forest plot for gender sub-groups analysis on NPV. A significant positive small effect was found with low levels of heterogeneity both in female (SMD, 0.48; 95% CI: 0.11 to 0.86; I^2^, 16%; Z, 2.51; *p* = 0.01) and male (SMD, 0.46; 95% CI: 0.29 to 0.64; I^2^, 0%; Z, 5.16; *p* < 0.0001) sub-groups.

#### 3.4.3. Competitive Level

Figure 4C shows the forest plot for the competitive level analysis on NPV. A significant positive small effect was evident in both elite (SMD, 0.47; 95% CI: 0.28 to 0.67; I^2^, 0%; Z, 4.76; *p* < 0.00001) and amateur (SMD, 0.47; 95% CI: 0.18 to 0.73; I^2^, 0%; Z, 3.21; *p* = 0.001) competitive level with low heterogeneity in the two sub-groups.

#### 3.4.4. Training Type

Figure 4D displays the forest plot for training type analysis on NPV. All the sub-groups showed low levels of heterogeneity (I^2^, >21%). Both plyometric training (SMD, 0.51; 95% CI: 0.30 to 0.72; I^2^, 21%; Z, 4.83; *p* < 0.0001) and mixed strength and conditioning interventions (SMD, 0.39; 95% CI: 0.11 to 0.68; I^2^, 0%; Z, 2.74; *p* = 0.006) reached a significant positive effect in comparison to endurance training (SMD, 0.43; 95% CI: −0.10 to 0.97; I^2^, 0%; Z, 1.60; *p* = 0.08) with moderate and small effects, respectively.

### 3.5. Effect on Neuromuscular Power Horizontal

Figure 5 displays the forest plot for NPH. Twenty-four trials were included for analysis [49,50,51,61,62,64,65,66,67,68,70,71,74,77,78] with a total of 254 participants. A significant overall moderate positive effect was found in NPH; however, moderate levels of heterogeneity were evident (SMD, −0.55; 95% CI: −0.88 to −0.22; I^2^, 68%; Z, 3.23; *p* = 0.001).

#### 3.5.1. Age

Figure 6A shows the forest plot for the age sub-groups analysis on NPH. Moderate to high level of heterogeneity was found in all the analysis except from the U-12 group with low levels (I^2^, 0%, *p* = 0.46). Significant large positive effect was found in the U-12 sub-groups (SMD, −1.38; 95% CI: −2.28 to −0.48; I^2^, 0%; Z, 3.01; *p* = 0.003). No significant effect was found either in the U-14 (SMD, −0.52; 95% CI: −1.43 to 0.43; I^2^, 84%; Z, 1.07; *p* = 0.29), U-16 (SMD, −0.42; 95% CI: −0.93 to 0.09; I^2^, 62%; Z, 1.63; *p* = 0.10) and U-18 sub-groups (SMD, −0.55; 95% CI: −1.11 to 0.01; I^2^, 68%; Z, 1.87; *p* = 0.06).

#### 3.5.2. Gender

Figure 6B displays the forest plot for the gender sub-groups analysis on NPH. High levels of heterogeneity were found in both sub-groups. A significant positive effect was evident both in female (SMD, −0.95; 95% CI: −1.90 to 0.00; I^2^, 73%; Z, 1.96; *p* = 0.05) and male sub-groups (SMD, −0.47; 95% CI: −0.83 to −0.12; I^2^, 67%; Z, 2.61; *p* = 0.009), with large and small effects, respectively.

#### 3.5.3. Competitive Level

Figure 6C shows the forest plot for competitive level sub-groups analysis on NPH. High levels of heterogeneity were evident in the elite sub-groups (I^2^, 73%, *p* < 0.00001), however, moderate levels were found in amateur (I^2^, 47%, *p* = 0.08). Both elite (SMD, −0.49; 95% CI: −0.93 to −0.06; I^2^, 73%; Z, 2.24; *p* = 0.03) and amateur (SMD, −0.70; 95% CI: −1.17 to −0.22; I^2^, 47%; Z, 2.86; *p* = 0.004) sub-groups reached significant positive effect with small and moderate effects, respectively.

#### 3.5.4. Training Type

Figure 6D displays the forest plot for training type analysis on NPH. Except from mixed strength and conditioning (I^2^, 0%), all the sub-groups showed high levels of heterogeneity (I^2^ < 75%). From all the training types, only plyometric training, with a moderate effect (SMD, −0.79; 95% CI: −1.40 to −0.18; I^2^, 78%; Z, 2.53; *p* = 0.01), reached significant levels in comparison to mixed strength and conditioning (SMD, −0.27; 95% CI: −0.59 to 0.06; I^2^, 0%; Z, 1.61; *p* = 0.11) and endurance training (SMD, −0.54; 95% CI: −1.33 to 0.25; I^2^, 75%; Z, 1.33; *p* = 0.18).

### 3.6. Effect on Endurance

Figure 7 illustrates de forest plot for endurance. Only 3 studies [64,65,79] were included in this analysis resulting in a significant positive large effect (SMD, 0.88; 95% CI: 0.17 to 1.58; I^2^, 50%; Z, 2.43; *p* = 0.02) with moderate levels of heterogeneity.

### 3.7. Meta-Regression

The main results from the meta-regression model can be found in Table 4 for NPV and Table 5 for NPH. Regression models are shown in Figure 8. In NPV, considering the models that reached the minimum number of studies, only U-16 age group model was able to significantly predict changes in jump performance with small accounted variance (Adjusted R^2^ = 0.78). Elite level model reached a significant prediction, nevertheless, moderate variability was accountable (Adjusted R^2^ = 0.57). Even though both Male and Plyometric training models were able to significantly predict jump performance, the high levels of variance accounted in the model limits its validity predicting NPV changes (Male; Adjusted R^2^ = 0.34, Plyometric training; Adjusted R^2^ = 0.36).

In NPH, only plyometric training, male and elite level models reached the minimum number of studies for meta-regression analysis. All models significantly predict multidirectional speed and agility with very small variance accounted (Plyometric training; Adjusted R^2^ = 0.84, Male; Adjusted R^2^ = 0.83, Elite; Adjusted R^2^ = 0.83).

## 4. Discussion

The main aim of this systematic review with meta-analysis and meta-regression was to collect, describe and analyze the effect of multidisciplinary neuromuscular and endurance interventions on performance in youth basketball players with respect to age, level, gender of the basketball players and according to the type of the intervention performed. Additionally, and considering the available data, to try to establish a prediction model on the changes in NPV and NPH considering these parameters through a meta-regression analysis. From the best of the author’s knowledge this is the first systematic review with meta-analysis and meta-regression focusing on this topic in youth basketball players.

Considering the data available in the literature and included in this study, the main results indicate that multidisciplinary interventions, including plyometric training, mixed strength and conditioning, HIIT basketball programs and repeated sprint training have an overall positive effect on jump (NPV), multidirectional speed and agility (NPH) and endurance (E). However, moderate to high variability in NPH (I^2^, 72%) and E (I^2^, 50%) outcomes were evident and may have an impact on the conclusions. Nonetheless, considering the moderate heterogeneity found in E, more studies are needed in order to achieve conclusive evidence.

Concerning the sub-groups analysis for age, the results in vertical jump show a significant increase only in U-16 and U-18. This is supported by meta-regression analysis with U-16 model being able to predict changes in vertical jump capacity. Even though more studies are needed in U-18, the model is able to significantly explain 50% of the changes in vertical jump performance. Conversely, the results in multidirectional speed and agility showed a significant improvement in multidirectional speed and agility only in U-12 group. This differs with the meta regression analysis suggesting a good predictability of the U-16 and U-18 with very low variance. Chronological age and maturation involves periods of accelerated growth and development related to neural and hormonal changes that maximize the development of specific capacities [38]. In particular, youth basketball players experience changes in their anthropometric and physiological parameters that promote performance changes throughout their individual development in different maturational stages [81]. However, it is important to consider that maturational changes will occur simultaneously with motor control coordination evolution. Consequently, those overlapped changes will limit the effectiveness in muscular contraction due to alterations in motor unit recruitment and inter and intra muscular coordination [38]. According to Long Term Athlete Development approach [82] and the Youth Physical Development Model [37], the peak development in strength and power appears 12 to 18 month after the peak high velocity (PHV) or from around 16 years old to adulthood in terms of chronological age [32,83]. In fact, Calleja-González et al. [81] pointed U-16 as the offset for peak changes in youth players from a top basketball academy. Our results correspond with the previous literature, increasing in vertical jump, only evident in the U-16 and the U-18 age group. This in fact highlights the importance of strength and power development at this maturational stages as game demands increase with age [84] and performance needs to be higher coinciding with the difficult transition from U-18 to senior in basketball [48,85]. For this reason, strength and conditioning programs should take advantage from this “sweet spot” to develop and prepare youth basketball players for future performance and in order to facilitate the senior transition and avoid injury risk [36,86,87].

Other studies suggest that strength levels were more developed in U-14 [32,88], however, muscular power development is more evident in the adolescence due to changes related to PHV [32,89,90].Those results support our findings, as according to the literature, vertical jump involves strength and speed during movement production and therefore muscular power [17,91]. Muscular power development post PHV might be attributed to hormonal changes, fiber type distribution and increase in muscle mass during adolescence [32,37,38]. Multiple interventions (in particular plyometric training) are a well-established method to improve speed in different stages from childhood to late maturity of an athlete [23,42,85,92,93,94,95]. According to the literature, in the early childhood, it seems to be primarily attributed to neural capacity, motor recruitment and coordination [33,42,96]. These changes and the high neural plasticity within the early stages could explain that only significant changes were only evident in the U-12 group. As mentioned, it is important to consider that body changes during puberty might alter coordination patterns and reduce sprint ability despite hormonal and muscle mass changes during this stage [35,97]. However, higher sprint and acceleration ability is found during PHV or post-PHV due to maturation-related changes within these stages [98,99,100]. This is not supported by the results from the meta-analysis. Nevertheless, assuming the low number of studies included in the meta-regression analysis, it suggests promising results as U-16 and U-18 age group models seem to predict multidirectional speed and agility changes with very low variance accounted.

Gender also had an impact on the results. Female and male improved similarly in both vertical jump and multidirectional speed and agility. Similar training loads are typically applied to male and female athletes to improve performance; although sex and physical differences exist between them [101,102,103]. According to the literature, it seems that muscle cross sectional area and pennation angle are more related to age, despite difference in gender can be found [104]. There is evidence showing that female athletes have a lower percentage of type I fibers as well as lower cross-sectional area in type II muscle fiber than youth male athletes [101,103,105]. This physiological and anatomical differences are associated with greater force generation and contractile velocity and as a result, with a greater power output in males [104,106]. Moreover, male youth athletes are capable of recruiting more motor units. This further contributes to the ability to create more force development and to better use the stretch-shortening cycle than female youth athletes and even more after PHV [107,108]. When comparing absolute values, it is evident that males register higher values in force, power, speed and agility than females [28,104,109,110]. However, considering relative values, this was not affected by gender [104]. For this reason, individual trainability according to gender must be considered. In this regard, Lesinski et al. [89] found similar resistance training related gains in muscle strength and vertical jump in both males and females, although few studies with female athletes were included. Research comparing youth adaptations considering gender has shown significantly greater resistance training-induced improvements in female youth athletes compared to their male counterparts. It is important to consider the lack of studies and information related to age groups included [32,87,89]. The results from the meta-analysis suggest that youth female basketball players are able to improve in a similar way than male. Additionally, considering the number of included studies (N = 5), the meta-regression analysis showed that female model seems to predict jump performance with very low variance, whereas male reached significant predictability. However, the high variance accounted alongside the reduced number of studies included limits the confidence on the model. Differently, male model seems to predict changes in multidirectional speed and agility with very low levels of variance whereas female did not reach the minimum number of studies to extract reliable conclusions. Hence, although different physiological changes related to age and maturation between gender exists, trainability of female youth basketball players may be at least similar compared with males [89]. However, more studies in female youth athletes could show a clearer picture and will help researchers to establish more adjusted prediction models.

Competitive level of the players showed that the different interventions performed in elite and amateur basketball players induced positive changes in jumping ability and in multidirectional speed and agility on both groups. Literature about youth basketball reflects a difference between amateur and elite athletes in terms of anthropometric measurements, performance outcomes and physiological capacities [111,112]. The results from this meta-analysis suggest that the trainability of the players is not related with the level and experience of the athlete. Both amateur and elite basketball players improved, in a similar way, their jumping ability performance and their multidirectional speed and agility with different interventions, although a moderate effect was evident in the amateur multidirectional speed and agility sub-groups. This is in line with the results from the meta-regression analysis as that suggests that elite and amateur models are able to predict changes in both jump performance and multidirectional speed and agility with moderate to small levels of variance accounted. It is important to note that even though the sample size for elite is high for both NPV (N = 19) and NPH (N = 16), more studies are needed in amateur athletes (N = 9) to have a higher confidence in the results of the meta-regression model to predict changes according to level of the participants. Other systematic reviews have found controversial results in jump performance [95,113], sprinting and change of direction [114] after multiple interventions. Slimani, Paravlić & Bragazzi [113] found a greater effect of plyometric training on vertical jump in elite athletes whilst Stojanović et al. [95] showed that amateur and non-elite athletes improved more effectively in vertical jump compared to elite athletes. This is most probably due to a higher response in the unexperienced athletes. However, the unbalanced comparison among studies (15 non-elite vs. 1 elite) and the fact that this review only included female athletes makes it difficult to compare to our findings. Considering the results in multidirectional speed, the results from Thapa et al. [114] indicate that elite athletes have greater response to training in 30 m linear sprint and change of direction. The authors attribute those results to the higher strength levels and more fast twitch muscle fibers of elite compared to amateur athletes [114]. Conversely, the highest positive effect in this meta-analysis was found in the amateur multidirectional speed and agility sub-groups. With respect to the available literature and the results from the meta-analysis and the meta-regression analysis, it is still unclear if trainability is linked to the level of the athletes. For this reason, trainability should be an important factor to consider in future studies as the interventions will be applied to youth basketball players with different levels and years of experience.

Concerning the type of the intervention used, the results show that plyometric training and mixed strength and conditioning interventions improve vertical jump, but only plyometric training could induce significant changes in multidirectional speed and agility. The results from the meta-regression analysis suggest that plyometric training could predict changes in multidirectional speed and agility with very low levels of variance. However, mixed strength training also seems to predict NPH changes with very low levels of variance accounted despite the low number of studies were included. Regarding vertical jump, the models suggest that both plyometric training and mixed strength and conditioning predict jump performance, although mixed strength and conditioning model did not include enough studies to provide precise conclusions. Additionally, high to moderate variance was found in the models.

Behm et al. [41], reported that force and velocity need to be stimulated in order to improve mechanical power since both are involved in the power equation. There is evidence that both strength training [89,115] and plyometric training [24,95,116] improve vertical jump in a similar way. However, the combination of both resistance and plyometric training could promote even higher improvements in jump capacity in youth [23,32,89,114,115,117,118]. Plyometric training increases rate of force development, muscular power and muscle contraction velocity through maximizing stiffness, mean cross sectional area and peak power in all muscle fibers [115,119]. On the other hand, resistance training is able to induce similar adaptions through muscle hypertrophy, increased myosin heavy chain IIa (MHC IIa), and changes in neural adaptations such us motor unit frequency and activation [115]. Regarding speed improvement, Behm et al. [41] found a higher improve applying resistance training compared to plyometric training in adolescent and children. Although a positive effect was also found when applying plyometric training. The results from other studies also support the finding on the meta-analysis suggesting that plyometric training is beneficial in order to improve multidirectional sprint and agility [21,22,23,25,26,120,121]. The benefits induced by plyometric training might be due to preferential motor unit recruitment, velocity of muscle contraction and transfer of energy during sprinting [122]. Although a positive effect of plyometric training on both vertical jump and multidirectional speed is evident, specific considerations regarding technique and training volume will be fundamental in order to avoid injury risk and high join impact on those development stages [119].

### 4.1. Perspective

The understanding on how to better develop youth basketball players avoiding injury risk is one of the biggest concerns for strength and conditioning coaches and sports scientist [16]. It is fundamental to identify guidelines for youth basketball development considering age, gender and level. This systematic review with meta-analysis and meta-regression suggests plyometric training as the best training method to improve jump capacity in youth basketball players, especially over U-16. Consequently, plyometric training in early adolescence should focus on establishing coordination patterns and motor control, avoiding high impacts on the joints, whereas in late adolescence power development should be the target corresponding with the window of opportunity in the development stage [33,35,37,41,42,123]. The inclusion of a mixed strength and conditioning program alongside plyometric training seems to maximize those benefits at this development stages. Moreover, it has a clear potential for injury prevention in young population in different sports, hence increasing the options for long-career development [36,86,87,124,125]. Post PHV stages will be really important for strength and power development and facilitate a better transition to the higher demands in senior level and avoid injury risk. As said before, the implementation of more endurance interventions in youth basketball players should be of special interest, considering the importance of this capacity in senior high-level basketball and the lack of studies performing this type of interventions in the literature. High trainability was observed both in male and female youth basketball players. Considering the suggested prediction on the female model despite the lack of existing studies, elucidates the need for more studies to better understand female basketball players development. Competitive level and experience of the youth athletes seems to be independent of trainability and thus, should be considered in training. Both elite and amateur athletes benefit from the different interventions and improved in a similar way in jump and multidirectional speed. More data considering the potential capacity for adaptation and injury risk related to the experience and level of the youth basketball players should be considered in future research for program adaptations and applications. Safety considerations must be taken throughout the whole youth basketball players development to avoid injury risk.

Finally, considering the heterogeneity found in the performance tests used throughout the different studies included, it is necessary to establish common guidelines for performance assessment to improve study designs, to better compare outcomes, to identify and correct player weaknesses or injury risks and eventually improve player development in the future. Recent systematic reviews described the most implemented physical fitness assessments in basketball [3,27]. Likewise, high heterogeneity in the tests used to assess performance in different basketball capacities was evident. These results highlight the need for consensus assessing physical fitness and performance in basketball, especially in youth.

### 4.2. Limitations

Some limitations were evident in the study and should be considered. High levels of heterogeneity were evident in the measurements related to NPH and E. I^2^ above 75% was found on the overall analysis and also in the sub-group analysis in the NPH, representing large levels of heterogeneity. This might be due to the different protocols used in the included studies. A random effect model was used to reduce the effects of this limitation [126]. In E overall analysis presented moderate levels of inconsistency with I^2^ equal to 50%. In this case, this might be related to the few studies included in E and highlights another limitation of this meta-analysis. This issue prevented to do sub-groups analysis and investigate the effect of the interventions on E in more detail.

The available data in the literature limits the conclusions of the meta-regression analysis. However, the results suggest open directions for the future including more studies in youth basketball population.

Only chronological age was considered for sub-groups analysis because insufficient data regarding maturational stage or PHV was available in the included studies. This limits the conclusions regarding age sub-groups as athletic development stages are correlated to biologic maturation and PHV but no to chronological age [33,37]. Additionally, the age sub-groups did not differ between male and female, despite differences in maturational development are evident between gender [101].

## 5. Conclusions

In conclusion, multidisciplinary interventions including plyometric training, mixed strength and conditioning, HIIT basketball programs and repeated sprint training are able to induce positive changes in the jump capacity, multidirectional speed and also in endurance in youth basketball players. Those capacities are key in performance in youth basketball and, therefore, it is fundamental to establish a progressive and careful training periodization of those capacities for future development and transition to senior level considering age, gender, and level to maximize performance. The data availability in the scientific literature including interventions based on gender and level of the basketballers and the lack of combination of different training interventions, elucidates the need for more studies to understand youth basketball development.

## Figures and Tables

**Figure 1 ijerph-19-09642-f001:**
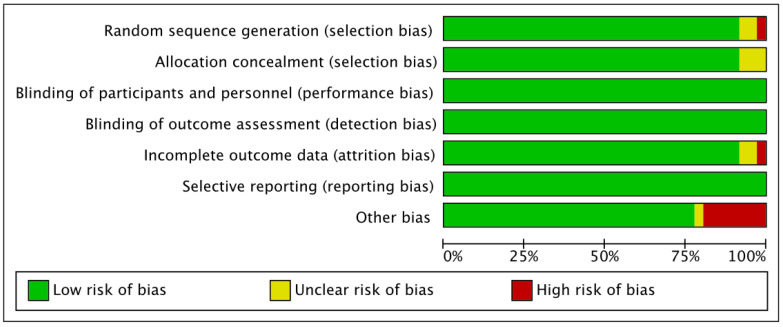
Risk of bias assessment summary.

**Figure 2 ijerph-19-09642-f002:**
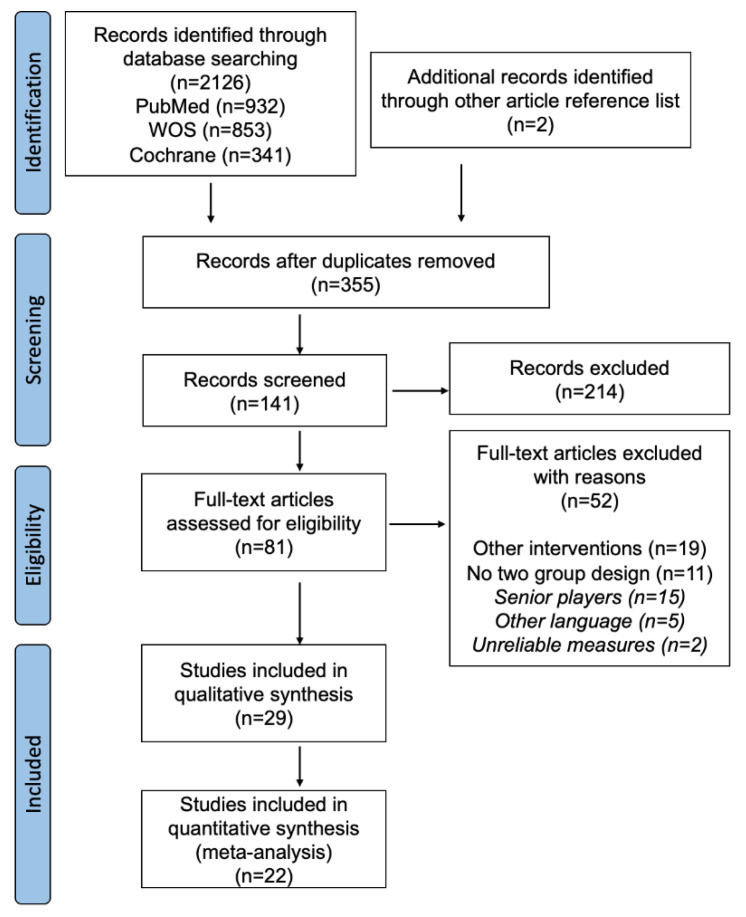
Flow diagram of study selection in PRISMA^®^ 35.

**Figure 3 ijerph-19-09642-f003:**
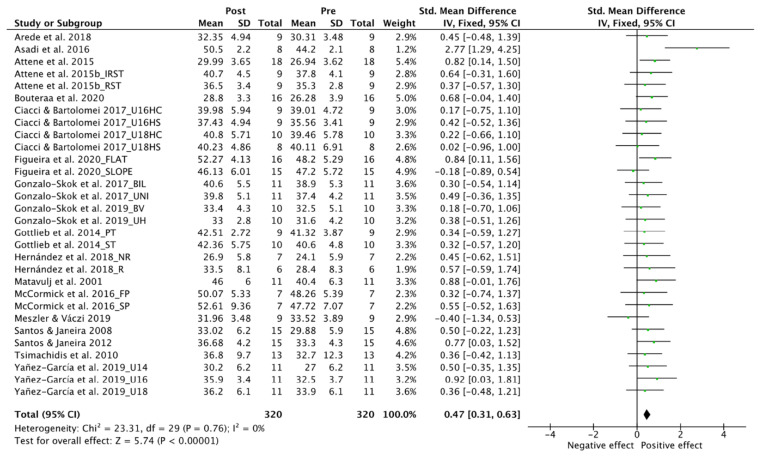
Forest Plot comparing the effect of multidisciplinary interventions on neuromuscular power vertical (NPV) [49,50,52,62,63,64,66,67,68,70,71,72,73,74,75,76,77,78].

**Figure 4 ijerph-19-09642-f004:**
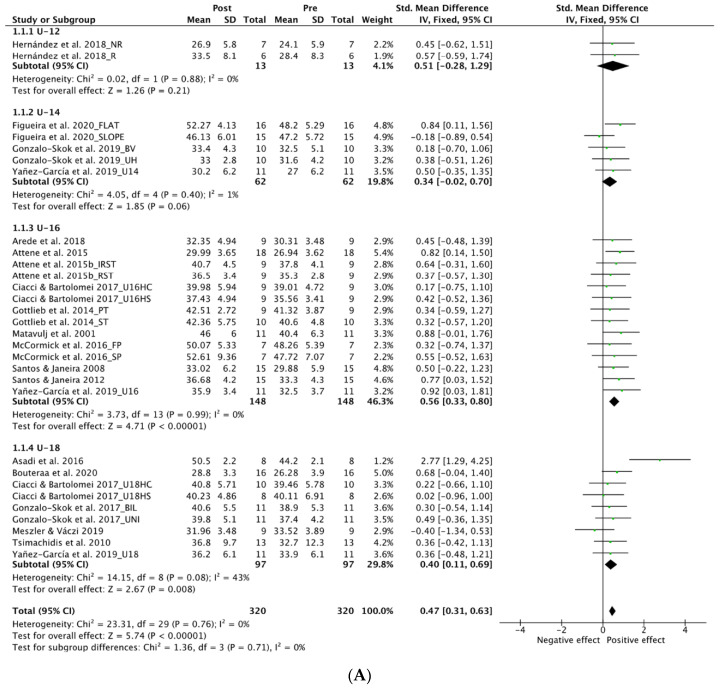
Forest Plot comparing the effect of multidisciplinary interventions on neuromuscular power vertical (NPV) considering sub-groups analysis for age (**A**), gender (**B**), level (**C**) and type of intervention (**D**) [49,50,52,62,63,64,66,67,68,70,71,72,73,74,75,76,77,78].

**Figure 5 ijerph-19-09642-f005:**
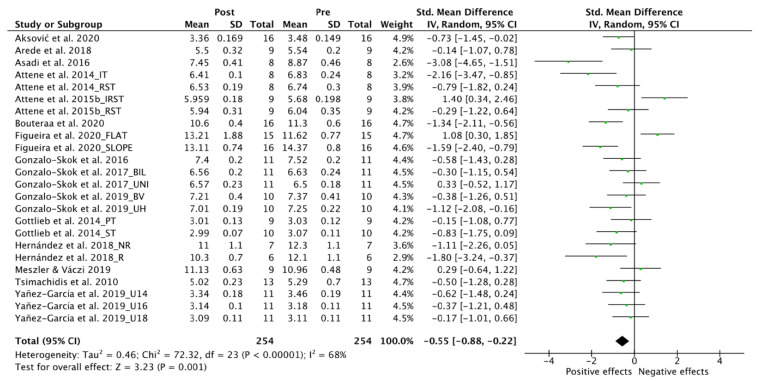
Forest Plot comparing the effect of multidisciplinary interventions on neuromuscular power horizontal (NPH) [49,50,51,61,62,64,65,66,67,68,70,71,74,77,78].

**Figure 6 ijerph-19-09642-f006:**
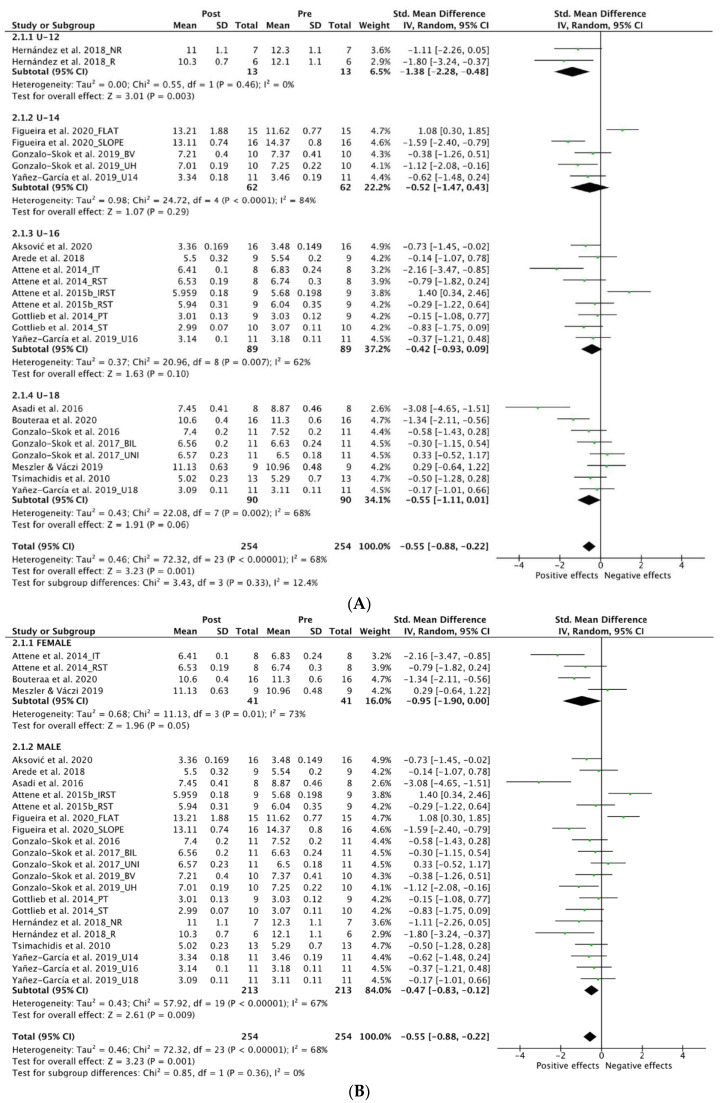
Forest Plot comparing the effect of multidisciplinary interventions on neuromuscular power horizontal (NPH) considering sub-groups analysis for age (**A**), gender (**B**), level (**C**) and type of intervention (**D**) [49,50,51,61,62,64,65,66,67,68,70,71,74,77,78].

**Figure 7 ijerph-19-09642-f007:**
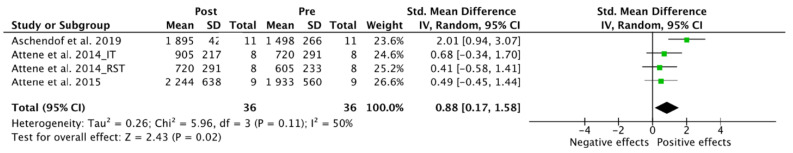
Forest Plot comparing the effect of multidisciplinary interventions on endurance [63,65,79].

**Figure 8 ijerph-19-09642-f008:**
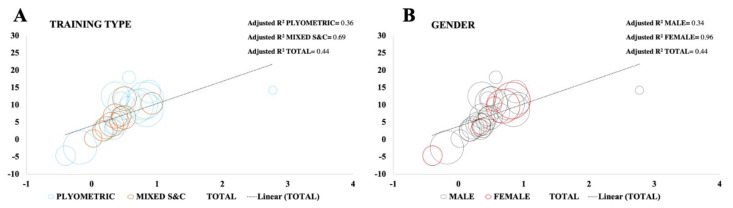
Meta regression models for neuromuscular power vertical (**A**–**D**) considering sub-groups: training type (**A**), gender (**B**), level (**C**) and age (**D**) and neuromuscular power horizontal (**E**–**H**) considering sub-groups: training type (**E**), gender (**F**)**,** level (**G**) and age (**H**).

**Table 1 ijerph-19-09642-t001:** PICOS model for inclusion criteria definition.

P (Population)	“youth basketball players”
I (Intervention)	“neuromuscular and endurance conditioning interventions”
C (Comparators)	“group comparison with multidisciplinary interventions and controls”
O (Outcomes)	“neuromuscular power (vertical and horizontal) and endurance”
S (Study design)	“any type of design”

**Table 2 ijerph-19-09642-t002:** Summary of the studies included in the meta-analysis. Bold numbers indicate significant changes between pre and post.

Author	Participants/ Gender	Age/ Level	Duration/Type of Intervention	Outcome	Pre Test (Unit): M ± SD	Post Test (Unit): M ± SD	Sig
Aksović et al. 2020 [61]	33, M	U-16 amateur	10 weeks. Plyometric program	NPH	S5m (s): 1.21 ± 0.034 S10m (s): 2.03 ± 0.048 S20m (s): 3.48 ± 0.149	**S5m (s): 1.14 ± 0.066 *** S10m (s): 1.95 ± 0.084 **S20m (s): 3.36 ± 0.169 ***	* significant main effect pre and post
Arede et al. 2018 [49]	16, M	U-16 amateur	8 weeks. Mixed strength and conditioning program	NPV NPH	CMJ (cm): 30.31 ± 3.48 SJ (cm): 27.24 ± 2.91 S10m (s): 2.3 ± 0.11 505 (s): 5.54 ± 0.20	**CMJ (cm): 32.34 ± 4.94 *†** **SJ (cm): 29.37 ±3.72 *** **S10m (s): 1.95 ± 0.07 *†** 505 (s): 5.50 ± 0.17	* significant pre and post † significant EG vs. CG
Asadi et al. 2016 [62]	16, M	U-18 elite	8 weeks. Plyometric training program.	NPV NPH	VJ (cm): 44.2 ± 2.1 BJ (cm): 228 ± 4.4 *t*-Test (s): 13.34 ± 0.50 Illinois (s): 18.92 ± 0.76 1RM (kg): 183 ± 8.7 S60m (s): 8.87 ± 0.46	**VJ (cm): 50.5 ± 2.2 *†** **BJ (cm): 239 ± 3.9 *†** ***t*-Test (s): 12.20 ± 0.32 *†** **Illinois (s): 17.81 ± 0.71 *†** **1RM (kg): 200 ± 9.8 *†** **S60m (s): 7.45 ± 0.41 *†**	* significant pre and post † significant EG vs. CG
Aschendorf et al. 2019 [79]	25, F	U-16 elite	5 weeks. B- specific HIIT program	E	Yo-Yo (m): 1498 ± 266 S20m (s): 5.62 ± 0.30 S20m ball (s): 5.79 ± 0.31 CMJ (cm): 26.85 ± 3.73 CMJ arm (cm): 29.95 ± 4.59 SJ (cm): 25.45 ± 3.43 LJ (cm): 177 ± 10.2 Pass (m): 9.85 ± 0.79	**Yo-Yo (m): 1895 ± 42 *** **S20m (s): 5.53 ± 0.22 *** **S20m ball (s): 5.72 ± 0.24 *** CMJ (cm): 27.05 ± 3.63 CMJ arm (cm): 30.52 ± 3.69 SJ (cm): 25.25 ± 3.69 LJ (cm): 177 ± 14.3 **Pass (m): 10.24 ± 0.58 ***†	* significant pre and post † significant EG vs. CG
Attene et al. 2014_RST [65]	16, F	U-16 elite	6 weeks. Repeated Sprint training	E	BT (s): 6.74 ± (0.3) WT (s): 7.26 ± (0.27) TT (s): 69.91 ± (2.73) BLa: 8.45 ± (2.41) FI%: 3.79 ± (1.57) Y Dis (m): 605 ± (233) Y Speed (Km·h): 14.5 ± (0.46) Y HR (bpm): 199 ± (8)	**BT (s): 6.53 ± (0.19) *** WT (s): 6.87 ± (0.27) TT (s): 67 ± (2.23) BLa: 7.11 ± (2.29) FI%: 2.69 ± (0.58) Y Dis (m): 775 ± (242) Y Speed (Km·h): 14.75 ± (0.38) Y HR (bpm): 200 ± (10)	* significant pre and post No significant main effect on training type
Attene et al. 2014_IT [65]	16, F	U-16 elite	6 weeks. Intermittent training	E	BT (s): 6.83 ± (0.24) WT (s): 7.28 ± (0.33) TT (s): 70.6 ± (2.42) BLa: 9.45 ± (1.63) FI%: 3.34 ± (1.34) Y Dis (m): 720 ± (291) Y Speed (Km·h): 14.81 ± (0.53) Y HR (bpm): 201 ± (5)	**BT (s): 6.41 ± (0.1) *** **WT (s): 6.89 ± (0.32) *** **TT (s): 66.37 ± (1.83) *** **BLa: 6.49 ± (1.35) *** FI%: 3.54 ± (1.87) **Y Dis (m): 905 ± (217) *** **Y Speed (Km·h): 14.94 ± (0.42) *** Y HR (bpm): 196 ± (6)	* significant pre and post No significant main effect on training type
Attene et al. 2015 [63]	36, F	U-16 elite	6 weeks. Plyometric training program.	NPV	CMJ Height (cm): 26.94 ± 3.62 CMJ power (w/kg): 24.52 ± 7.35 CMJ strength (n/kg): 20.22 ± 2.88 CMJ Speed (cm/s): 149 ± 31.06 SJ Height (cm): 22.71 ± 3.24 SJ power (w/kg): 29.64 ± 4.14 SJ max power (w/kg): 31.59 ± 3.87 SJ strength (n/kg): 20.76 ± 4.53 SJ Speed (cm/s): 183 ± 15.60	**CMJ Height (cm): 29.99 ± 3.65 *†** **CMJ power (w/kg): 27.47 ± 7.19 *** **CMJ strength (n/kg): 22.29 ± 2.60 *** **CMJ Speed (cm/s): 168 ± 28.03 *** **SJ Height (cm): 26.21 ± 3.55 *†** SJ power (w/kg): 31.77 ± 4.13 * SJ max power (w/kg): 32.21 ± 4.21 SJ strength (n/kg): 22.44 ± 2.98 **SJ Speed (cm/s): 197 ± 14.39 ***	* significant pre and post † just significant main effect for SJ and CMJ height in BPT group
Attene et al. 2015b_RST [64]	18, M	U-16 elite	4 weeks. Repeated Sprint training	E	RSA BT (s): 6.042 ± 0.351 RSA WT (s): 6.583 ± 0.418 RSA TT (s): 63.06 ± 3.731 RSA FI (%):4.4 ± 2.3 IRSA BT (s): 7.320 ± 0.463 IRSA WT (s): 7.888 ± 0.548 IRSA TT (s): 76.26 ± 5.095 IRSA FI (%): 4.2 ± 1.2 SJ (cm): 35.0 ± 4.0 CMJ (cm): 35.3 ± 2.8 Yo-Yo (m): 1233 ± 663	RSA BT (s): 5.940 ± 0.307 * RSA WT (s): 6.277 ± 0.410 * RSA TT (s): 61.24 ± 3.593 RSA FI (%): 3.1 ± 1.3 * IRSA BT (s): 7.206 ± 0.439 * IRSA WT (s): 7.565 ± 0.436 IRSA TT (s): 73.92 ± 4.527 IRSA FI (%): 2.6 ± 0.9 SJ (cm): 36.1 ± 3.2 CMJ (cm): 36.5 ± 3.4 * Yo-Yo (m): 1549 ± 679 *	* significant pre and post
Attene et al. 2015b_IRST [64]	18, M	U-16 elite	4 weeks. Intensive Repeated Sprint training	E	RSA BT (s): 5.676 ± 0.198 RSA WT (s): 6.121 ± 0.280 RSA TT (s): 58.89 ± 2.040 RSA FI (%): 3.8 ± 1.4 IRSA BT (s): 6.877 ± 0.267 IRSA WT (s): 7.385 ± 0.360 IRSA TT (s): 71.25 ± 2.943 IRSA FI (%): 3.6 ± 1.0 SJ (cm): 39.3 ± 3.6 CMJ (cm): 37.8 ± 4.1 Yo-Yo (m): 1933 ± 560	RSA BT (s): 5.595 ± 0.180 RSA WT (s): 5.975 ± 0.342 RSA TT (s): 57.57 ± 2.163 **RSA FI (%): 2.9 ± 1.2 *** **IRSA BT (s): 6.665 ± 0.245 *** **IRSA WT (s): 7.011 ± 0.327 *** IRSA TT (s): 68.54 ± 2.986 IRSA FI (%): 2.8 ± 1.5 **SJ (cm): 41.3** **±** **3.5 *** **CMJ (cm): 40.7** **±** **4.5 *†** Yo-Yo (m): 2244 ± 638	* significant pre and post † just significant main effect for CMJ in IRSAG
Bouteraa et al. 2020 [66]	22, F	U-18, amateur	8 weeks. Plyometric training program.	NPV NPH	SJ (cm): 20.4 ± 3.9 CMJ (cm): 26.8 ± 3.8 DJ (cm): 24.7 ± 2.9 DJ (w/kg): 26.5 ± 4.0 S5m (s): 0.95 ± 0.08 S10m (s): 1.82 ± 0.14 S20m (s): 3.39 ± 0.25 SBT (cm): 17.3 ± 9.6 YBT (cm): 104.3 ± 9.7 Illiniois (s): 11.3 ± 0.6	SJ (cm): 22.5 ± 3.5 CMJ (cm): 28.8 ± 3.3 **DJ (cm): 28.4 ± 3.0 *†** DJ (w/kg): 27.8 ± 5.0 S5m (s): 0.91 ± 0.05 S10m (s): 1.72 ± 0.09 S20m (s): 3.27 ± 0.16 **SBT (cm): 39.2 ± 13.3 *†** **YBT (cm): 114.4 ± 10.4 *†** **Illiniois (s): 10.6 ± 0.4 *†**	* significant pre and post † just significant main effect for EG compared to CG
Figueira et al. 2020_FLAT [67]	31, M	U-14 elite	4 weeks. Plyometric training program flat	NPV NPH	Standing height jump (cm): 48.20 ± 5.29 Drop Jump (cm): 42.00 ± 7.44 Contact time (ms): 200.20 ± 26.62 Jump Power (W): 1171.20 ± 256.01 Jump power (W/kg): 20.95 ± 2.90 AnAl Power (W): 859.33 ± 160.16 AnAl Power (Wkg): 15.36 ± 1.33 Hexagon agility (s): 13.21 ± 1.88	**Standing height jump (cm): 52.27 ± 4.13 †** Drop Jump (cm): 46.00 ± 7.51 Contact time (ms): 192.07 ± 26.95 Jump Power (W): 1331.33 ± 304.19 Jump power (wKg): 23.78 ± 4.66 AnAl Power (W): 882.00 ± 196.76 AnAl Power (Wkg): 15.59 ± 1.312 Hexagon agility (s): 11.62 ± 0.77	† just significant main effect for standing height jump in FLAT No look at pre to post changes
Figueira et al. 2020_SLOPE [67]	31, M	U-14 elite	4 weeks. Plyometric training program slope	NPV NPH	Standing height jump (cm): 47.20 ± 5.72 Drop Jump (cm): 42.20 ± 6.17 Contact time (ms): 238.20 ± 35.22 Jump Power (W): 923.07 ± 250.73 Jump power (Wkg): 17.56 ± 2.61 AnAl Power (W): 714.27 ± 195.70 AnAl power (Wkg): 13.50 ± 1.39 Hexagon agility (s): 14.37 ± 0.80	Standing height jump (cm): 46.13 ± 6.01 Drop Jump (cm): 43.53 ± 9.01 Contact time (ms): 216.80 ± 41.85 Jump Power (W): 1011.47 ± 225.18 Jump power (wKg): 19.30 ± 3.22 AnAl Power (W): 750.80 ± 204.46 AnAl power (Wkg): 14.16 ± 1.37 Hexagon agility (s): 13.11 ± 0.74	No look at pre to post changes
Gonzalo-Skok et al. 2017_BIL [50]	22, M	U-16 to U-18 elite	6 weeks. Mixed strength and conditioning program bilateral	NPV NPH	V-cut test (s): 6.63 ± 0.24 180° RCOD (s): 3.50 ± 0.12 180° LCOD (s): 3.48 ± 0.12 MP BIL (W): 407.5 ± 56.2 MP UNI R (W): 270.1 ± 32.8 MP UNI L (W): 270.9 ± 51.3 BL Imb (%): 6.9 ± 5.0 BL Def (%): 23.1 ± 8.4 S5m (s): 1.10 ± 0.05 S15m (s): 2.52 ± 0.09 S25m (s): 3.80 ± 0.15 CMJ (cm): 38.9 ± 5.3	V-cut test (s): 6.56 ± 0.20 180° RCOD (s): 3.45 ± 0.09 180° LCOD (s): 3.48 ± 0.11 **MP BIL (W): 470.2 ± 75.8 **** **MP UNI R (W): 330.1 ± 61.2 *** **MP UNI L (W): 330.4 ± 73.8 *** BL Imb (%): 4.4 ± 2.5 BL Def (%): 27.0 ± 6.4 **S5m (s): 1.06 ± 0.02 *** **S15m (s): 2.46 ± 0.07 *** **S25m (s): 3.69 ± 0.15 *** CMJ (cm): 40.6 ± 5.5	* Very likely changes pre and post ** almost certainly changes pre to post
Gonzalo-Skok et al. 2017_UNI [50]	22, M	U-16 to U-18 elite	6 weeks. Mixed strength and conditioning program unilateral	NPV NPH	V-cut test (s): 6.57 ± 0.23 180° RCOD (s): 3.54 ± 0.15 180° LCOD (s): 3.55 ± 0.17 MP BIL (W): 406.9 ± 48.5 MP UNI R (W): 273.7 ± 40.0 MP UNI L (W): 271.2 ± 36.8 BLImb (%): 9.6 ± 3.8 BLDef (%): 24.8 ± 8.9 S5m (s): 1.13 ± 0.05 S15m (s): 2.54 ± 0.08 S25m (s): 3.84 ± 0.12 CMJ (cm): 37.4 ± 4.2	V-cut test (s): 6.50 ± 0.18 **180° RCOD (s): 3.47 ± 0.10 *** **180° LCOD (s): 3.46 ± 0.15 †** **MP BIL (W): 466.3 ± 72.8 **** **MP UNI R (W): 372.2 ± 57.3 **†** **MP UNI L (W): 370.4 ± 50.4 **†** **BLImb (%): 4.8 ± 1.3 *†** BLDef (%): 36.8 ± 7.7 **S5m (s): 1.07 ± 0.06 **** **S15m (s): 2.48 ± 0.1 **** **S25m (s): 3.75 ± 0.15 **** **CMJ (cm): 39.8 ± 5.1 ***	* Very likely changes pre and post ** almost certainly changes pre to post † likely changes between UNI and BIL
Gonzalo-Skok et al. 2016 [51]	22, M	U-16 to U-18 elite	6 weeks. Mixed strength and conditioning program repeated power training	NPH	RSA BT (s): 7.16 ± 0.23 RSA ST (s): 7.86 ± 0.29 RSA M (s): 7.52 ± 0.2 %DEC RSA(%): 5.1 ± 1.8 RCOD BT (s): 6.58 ± 0.21 RCOD ST (s): 6.77 ± 0.20 RCOD MT (s): 6.86 ± 0.25 %DEC RCOD(%): 2.0 ± 0.7 UNIR (cm): 169.1 ± 16.8 UNIL (cm): 170.4 ± 16.6 LSIuni (%): 94.3 ± 3.6 DJR (cm): 402.2 ± 35.2 DJL (cm): 410.8 ± 24.5 LSIdj (%): 94.5 ± 4.4 LSIuni (%): 94.9 ± 4.4 DJR (cm): 394.5 ± 24.4 DJL (cm): 412.6 ± 17.8 LSIdj (%): 95.2 ± 3.4	RSA BT (s): 7.10 ± 0.18 **RSA ST (s): 7.67 ± 0.29 †** **RSA M (s): 7.40 ± 0.2 **†** %DEC RSA(%): 4.3 ± 1.7 **RCOD BT (s): 6.41 ± 0.20 **††** **RCOD ST (s): 6.61 ± 0.21 *††** **RCOD MT (s): 6.72 ± 0.23 *††** %DEC RCOD(%): 2.6 ± 1.5 **UNIR (cm): 180.9 ± 14.4 *†** **UNIL (cm): 182.7 ± 12.8 **†** LSIuni (%): 95.9 ± 2.3 DJR (cm): 411.1 ± 30.8 DJL (cm): 419 ± 28.9 LSIdj (%): 96.6 ± 2.3 LSIuni (%): 95.5 ± 4.3 DJR (cm): 393.4 ± 24.2 DJL (cm): 410.4 ± 14.3 LSIdj (%): 95.5 ± 4.6	* Very likely changes pre and post ** almost certainly changes pre to post † likely changes between RPA and CG †† very likely changes between RPA and CG
Gonzalo-Skok et al. 2019_UH [68]	20, M	U-14 elite	6 weeks. Plyometric training program unilateral horizontal	NPV NPH	S5m (s): 1.13 ± 0.07 S10m (s): 1.92 ± 0.08 S25m (s): 4.02 ± 0.20 CMJ (cm): 31.6 ± 4.2 CMJ L (cm): 12.9 ± 3.1 CMJ R (cm): 12.5 ± 3.0 HJ L (cm): 147.0 ± 22.5 HJ R (cm): 146.9 ± 19.0 V-cut (s): 7.25 ± 0.22 COD 180 (s): 2.72 ± 0.05 DORS L (cm): 10.5 ± 2.4 DORS R (cm): 10.0 ± 2.9 SEBT AL (cm): 55.9 ± 6.4 SEBT AR (cm): 53.7 ± 5.7 SEBT PLL (cm): 72.8 ± 8.4 SEBT PLR (cm): 72.3 ± 11.9	**S5m (s): 1.07 ± 0.05 **** **S10m (s): 1.86 ± 0.06 **†** S25m (s): 3.95 ± 0.19 CMJ (cm): 33.0 ± 2.8 **CMJ L (cm): 14.9 ± 1.9 *** **CMJ R (cm): 14.1 ± 2.9 *** **HJ L (cm): 159.0 ± 21.1 *** HJ R (cm): 153.2 ± 16.6 **V-cut (s): 7.01 ± 0.19 **** COD 180 (s): 2.72 ± 0.07 DORS L (cm): 10.4 ± 4.2 DORS R (cm): 10.6 ± 3.9 **SEBT AL (cm): 56.2 ± 6.8 *** SEBT AR (cm): 54.5 ± 7.3 SEBT PLL (cm): 74.9 ± 7.4 SEBT PLR (cm): 78.2 ± 11.6	* Very likely changes pre and post ** almost certainly changes pre to post † likely changes between BV and UH, more in UH
Gonzalo-Skok et al. 2019_BV [68]	20, M	U-14 elite	6 weeks. Plyometric training program bilateral vertical	NPV NPH	S5m (s): 1.14 ± 0.08 S10m (s): 1.91 ± 0.10 S25m (s): 3.99 ± 0.22 CMJ (cm): 32.5 ± 5.1 CMJ L (cm): 12.4 ± 3.6 CMJ R (cm): 12.3 ± 2.4 HJ L (cm): 141.0 ± 24.9 HJ R (cm): 143.2 ± 21.7 V-cut (s): 7.37 ± 0.41 COD 180 (s): 2.79 ± 0.17 DORS L (cm): 9.1 ± 2.7 DORS R (cm): 10.1 ± 2.4 SEBT AL (cm): 53.4 ± 6.3 SEBT AR (cm): 54.3 ± 5.2 SEBT PLL (cm): 69.3 ± 7.6 SEBT PLR (cm): 68.0 ± 7.8	S5m (s): 1.11 ± 0.06 S10m (s): 1.90 ± 0.10 S25m (s): 3.96 ± 0.21 CMJ (cm): 33.4 ± 4.3 CMJ L (cm): 14.1 ± 3.4 **CMJ R (cm): 14.6 ± 3.1 **** HJ L (cm): 152.8 ± 16.5 **HJ R (cm): 155.6 ± 17.9 *** V-cut (s): 7.21 ± 0.40 COD 180 (s): 2.77 ± 0.16 DORS L (cm): 9.6 ± 2.7 DORS R (cm): 10.0 ± 1.7 SEBT AL (cm): 56.1 ± 6.9 SEBT AR (cm): 55.6 ± 6.8 SEBT PLL (cm): 71.8 ± 7.3 SEBT PLR (cm): 72.9 ± 9.7	* Very likely changes pre and post ** almost certainly changes pre to post
Gottlieb et al. 2014_PT [70]	9,M	U-16 elite	6 weeks Plyometric training program	NPV NPH	CMJ (cm): 41.32 ± 3.87 S20m (s): 3.03 ± 0.12	CMJ (cm): 42.51 ± 2.72 S20m (s): 3.01 ± 0.13	
Gottlieb et al. 2014_ST [70]	10,M	U-16 elite	6 weeks Repeated sprint training program	NPV NPH	CMJ (cm): 40.60 ± 4.80 S20m (s): 3.07 ± 0.11	CMJ (cm): 42.36 ± 5.75 **S20m (s): 2.99 ± 0.07 ***	* significant pre and post
Matavulj et al. 2001 [72]	33, M	U-16 elite	6 weeks. Plyometric training program drop jumps	NPV	CMJ (cm): 40,4 ± 6.3	**CMJ (cm): 46± 6 *†**	* significant pre and post † more than control
McCormick et al. 2016_SP [73]	14, F	U-16 amateur	6 weeks. Plyometric training program sagittal plane	NPV	CMJ (cm): 47.72 ± 7.07 SLJ (cm): 177.89 ± 30.07 LHr (cm): 135.89 ± 22.36 LHl (cm): 140.06 ± 25.81 LSHr: 23.86 ± 3.13 LSHl: 24.00 ± 3.06	**CMJ (cm): 52.61 ± 9.36 *†** **SLJ (cm): 191.95 ± 29.06 *** **LHr (cm): 143.87 ± 25.34 *** **LHl (cm): 142.60 ± 32.33 *** **LSHr: 24.57 ± 2.99 *** **LSHl: 24.14 ± 2.55 ***	* significant pre and post † more than FPP
McCormick et al. 2016_FP [73]	14, F	U-16 amateur	6 weeks. Plyometric training program frontal plane	NPV	CMJ (cm): 48.26 ± 5.39 SLJ (cm): 176.89 ± 18.47 LHr (cm): 141.06 ± 7.47 LHl (cm): 137.16 ± 12.97 LSHr: 23.00 ± 2.31 LSHl: 22.71 ± 2.22	**CMJ (cm): 50.07 ± 5.33 *** **SLJ (cm): 187.05 ± 14.19 *** **LHr (cm): 154.94 ± 13.03 *** **LHl (cm): 153.49 ± 6.02 *Ψ** **LSHr: 24.57 ± 1.90 *** **LSHl: 24.71 ± 2.36 *Ψ**	* significant pre and post Ψ more than SPP
Meszler & Váczi (2019) [74]	16, F	U-18 amateur	7 weeks. Plyometric training program	NPV NPH	IAT (s): 16.21 ± 0.81 *t*-Test (s): 10.96 ± 0.48 CMJ (cm): 33.52 ± 3.89 B: 74.82 ± 2.22 MVC60ext (Nm): 165.64 ± 22.53 MVC60flex (Nm): 94.54 ± 14.05 MVC180ext (Nm): 120.25 ± 25.89 MVC180flex (Nm): 94.82 ± 24.99 H:Q60 ratio (%): 60.42 ± 9.99 H:Q180 ratio (%): 78.52 ± 10.32	**IAT (s): 16.95 ± 1.07 *** *t*-Test (s): 11.13 ± 0.63 **CMJ (cm): 31.96 ± 3.48 *** B: 75.62 ± 4.31 MVC60ext (Nm): 175.16 ± 21.61 MVC60flex (Nm): 99.11 ± 17.96 MVC180ext (Nm): 130.01 ± 19.06 **MVC180flex (Nm): 113.01 ± 26.80 *** H:Q60 ratio (%): 58.79 ± 7.72 H:Q180 ratio (%): 86.38 ± 10.51	* significant pre and post
Hernández et al. 2018_R [71]	19, M	U-12 amateur	7 weeks. Plyometric training program randomized	NPV NPH	CMJ (cm): 28.4 ± 8.3 DJ (cm): 20.6 ± 5.1 S30m (s): 5.71 ± 0.46 S30m ball (s): 7.18 ± 1.1 *t*-Test (s): 12.1 ± 1.1	**CMJ (cm): 33.5 ± 8.1 *†** **DJ (cm): 25.4 ± 5.9 *†** **S30m (s): 5.06 ± 0.52 *†** **S30m ball (s): 6.52 ± 1.0 *†** ***t*-Test (s): 10.3 ± 0.7 *†**	* significant pre and post † more RG group than the others
Hernández et al. 2018_NR [71]	19, M	U-12 amateur	7 weeks. Plyometric training program non randomized	NPV NPH	CMJ (cm): 24.1 ± 5.9 DJ (cm): 19.6 ± 6.5 S30m (s): 5.87 ± 0.51 S30m ball (s): 7.57 ± 1.7 *t*-Test (s): 12.3 ± 1.1	**CMJ (cm): 26.9 ± 5.8 *** **DJ (cm): 22.0 ± 6.0 *** **S30m (s): 5.48 ± 0.56 *** **S30m ball (s): 6.92 ± 1.6 *** ***t*-Test (s): 11.0 ± 1.1 ***	* significant pre and post
Santos & Janeira (2008) [75]	25, M	U-16 amateur	10 weeks. Plyometric training program	NPV	SJ (cm): 24.79 ± 4.2 CMJ (cm): 29.88 ± 5.9 ABA (cm): 34.77 ± 6.3 DJ (cm): 34.71 ± 7.4 MP (W·Kg): 23.69 ± 4.0 MBT (m): 3.47 ± 0.6	**SJ (cm): 28.01** **±** **4.6 *†** **CMJ (cm): 33.02** **±** **6.2 *†** **ABA (cm): 38.43** **±** **7.1 *** **DJ (cm): 36.64 ± 8.1 †** **MP (W·Kg): 24.48 ± 3.9** **MBT (m): 4.15 ± 0.5 *†**	* significant pre and post † significant different post training EG group than the CG
Santos & Janeira (2012) [76]	25, M	U-16 amateur	10 weeks. Plyometric training program	NPV	SJ (cm): 24.81 ± 3.3 CMJ (cm): 33.30 ± 4.3 ABA (cm): 38.73 ± 4.9 DJ (cm): 34.80 ± 4.1 MBT (m): 3.42 ± 0.38	**SJ (cm): 27.92** **±** **4.0 *†** **CMJ (cm): 36.68** **±** **4.2 *†** **ABA (cm): 42.62** **±** **4.4 *†** **DJ (cm): 38.10 ± 4.3 *†** **MBT (m): 3.68 ± 0.42 ***	* significant pre and post † significant different post training EG group than the CG
Ciacci & Bartolomei (2017)_U-16HS [52]	36, M	U-16 elite	16 weeks. Mixed strength and conditioning program half squat	NPV	U-16 SJ (cm): 34.19 ± 3.32 U-16 CMJ (cm): 35.56 ± 3.41 U-16 CMJarm (cm): 43.33 ± 5.47 U-16 TCMJstep (cm): 40.83 ± 4.98 U-16 Power Balance (%): 84.91 ± 9.62	**U-16 SJ (cm): 36.66 ± 3.34 *†** U-16 CMJ (cm): 37.43 ± 4.94 U-16 CMJarm (cm): 43.02 ± 4.46 U-16 **TCMJstep (cm): 43.94 ± 4.54 *†** U-16 Power Balance (%): 82.92 ± 7.01	* significant pre and post † significant different post training HSQ group than the HCL
Ciacci & Bartolomei (2017)_U-16HC [52]	36, M	U-16 elite	16 weeks. Mixed strength and conditioning program hang clean	NPV	U-16 SJ (cm): 37.32 ± 3.99 U-16 CMJ (cm): 39.01 ± 4.72 U-16 CMJarm (cm): 43.37 ± 5.82 U-16 TCMJstep (cm): 48.86 ± 4.24 U-16 Power Balance (%): 91.31 ± 8.37	U-16 SJ (cm): 37.16 ± 4.32 U-16 CMJ (cm): 39.98 ± 5.94 U-16 CMJarm (cm): 43.79 ± 6.82 U-16 TCMJstep (cm): 48.81 ± 6.75 U-16 Power Balance (%): 80.42 ± 14.63	post training HSQ group than the HCL
Ciacci & Bartolomei (2017)_U-18HS [52]	36, M	U-18 elite	16 weeks. Mixed strength and conditioning program half squat	NPV	U-18 SJ (cm): 38.44 ± 5.98 U-18 CMJ (cm): 40.11 ± 6.91 U-18 CMJarm (cm): 43.13 ± 7.14 U-18 TCMJstep (cm): 47.45 ± 7.86 U-18 Power Balance (%): 86.46 ± 7.98	**U-18 SJ (cm): 40.79 ± 4.97 *** U-18 CMJ (cm): 40.23 ± 4.86 U-18 CMJarm (cm): 46.71 ± 5.68 **U-18 TCMJstep (cm): 48.90 ± 5.5 *** U-18 Power Balance (%): 78.47 ± 9.05	* significant pre and post
Ciacci & Bartolomei (2017)_U-18HC [52]	36, M	U-18 elite	16 weeks. Mixed strength and conditioning program hang clean	NPV	U-18 SJ (cm): 38.78 ± 5.87 U-18 CMJ (cm): 39.46 ± 5.78 U-18 CMJarm (cm): 45.72 ± 6.58 U-18 TCMJstep (cm): 47.84 ± 7.71 U-18 Power Balance (%): 76.68 ± 7.62	**U-18 SJ (cm): 40.71 ± 5.17 *** **U-18 CMJ (cm): 40.80 ± 5.71 *** **U-18 CMJarm (cm): 47.33 ± 6.16 *** **U-18 TCMJstep (cm): 50.02 ± 6.96 *** U-18 Power Balance (%): 72.62 ± 10.30	* significant pre and post
Tsimachidis et al. 2010 [77]	26, M	U-18 amateur	10 weeks. Plyometric training program	NPV NPH	S30m (s): 5.29 ± 0.70 CMJ (cm): 32.7 ± 12.3	**S30m (s): 5.02 ± 0.23 *†** **CMJ (cm): 36.8 ± 9.7 *†**	* significant pre and post † significant different post training CTG than the CG
Yañez-García et al. 2019_U-14 [78]	11,M	U-14 elite	6 weeks Mixed strength and conditioning program	NPV NPH	S10m (s): 1.96 ± 0.12 S20m (s): 3.46 ± 0.19 CMJ (cm): 27.0 ± 6.2	**S10m (s): 1.88 ± 0.09 *** **S20m (s): 3.34 ± 0.18 *** **CMJ (cm): 30.2 ± 6.2 ***	* significant intra groups
Yañez-García et al. 2019_U-16 [78]	11,M	U-16 elite	6 weeks Mixed strength and conditioning program	NPV NPH	S10m (s): 1.83 ± 0.06 S20m (s): 3.18 ± 0.11 CMJ (cm): 32.5 ± 3.7	**S10m (s): 1.82 ± 0.06 *** **S20m (s): 3.14 ± 0.10 *** **CMJ (cm): 35.9 ± 3.4 ***	* significant intra groups
Yañez-García et al. 2019_U-18 [78]	11,M	U-18 elite	6 weeks Mixed strength and conditioning program	NPV NPH	S10m (s): 1.78 ± 0.07 S20m (s): 3.11 ± 0.11 CMJ (cm): 33.9 ± 6.1	**S10m (s): 1.78 ± 0.05 *** **S20m (s): 3.09 ± 0.11 *** CMJ (cm): 36.2 ± 6.1	* significant intra groups

% = percentage; 1RM = 1 repetition maximum; A = Anterior; ABA = abalakov; AnAl = Anaerobic Alactic; B = Balance; BJ = broad jump; BLa = Blood lactate; BLDef = bilateral deficit; BPT = basketball plyometric training group; BT = best time; BV = bilateral vertical group; CG = control group; cm = centimeters; CMJ = counter movement jump; CMJarm = counter movement jump arm use; COD = change of direction; CTG = combined training program group; Def = Deficit; DJ = drop jump; DORS = dorsiflexion test; EG = experimental group; F = female; FFP = frontal plane plyometric group; FI = fatigue index; FLAT = flat surface group; H:Q = hamstrings to quadriceps ratio; HCL = hang clean group; HJ = horizontal jump; HSQ = half squat group; IAT = Illinois agility test; Imb = Imbalance; IRSA = intermittent repeated sprint ability; IRSAG = Intermittent repeated sprint ability group; L = left; LH = lateral hop; LJ = long jump; LS = lateral shuffle; LSI = Limb symetry index; LSI = limb symmetry index; M = male; m = meters; MBT = medicine ball throw; min = minutes; MP = mechanical power; MPBIL = maximal power bilateral; MPUNI = maximal power unilateral; ms = miliseconds; MT = mean time; MVC = maximal voluntary contraction; NPH = neuromuscular power horizontal; NPV = neuromuscular power vertical; P = Posterior; PL = postero lateral; R = Right; RCOD = Repeated change of direction; RCODA = repeated change of direction ability; RG = randomized exercises group; RPA = repeated power ability group; RSA = repeated sprint ability; s = seconds; S10m = Sprint 10 m; S15m = Sprint 15 m; S20m = Sprint 20 m; S25m = Sprint 25 m; S30m = Sprint 30 m; S5m = Sprint 5 m; S60m = Sprint 60 m; SBT = stock balance test; SEBT = star excursion balance test; SJ = squat jump; SLJ = standing long jump; SLJ = single leg jump; TCMJ = counter movement jump with step approach; TCMJstep = Counter movement jump arm use with 1 step approach ;TT = total time; UH = unilateral horizontal group; UNI = Unilateral hop test; VJ = vertical jump; w/kg = wattage per weight in kilograms; w = wattage; WT = worst time; Y Dis = Yo-yo distance; Y HR = Yo-yo heart rate; Y time = Yo-yo time; YBT = Y balance test.

**Table 3 ijerph-19-09642-t003:** Number studies included in the meta-analysis per sub-group and references.

Competitive level of participants	Amateur	9 [49,61,66,71,73,74,75,76,77]
	Elite	13 [50,51,52,62,63,64,65,67,68,70,72,78,79]
Age	U-12	1 [71]
	U-14	3 [50,67,78]
	U-16	13 [49,52,61,63,64,65,70,72,73,75,76,78,79]
	U-18	7 [51,52,62,66,74,77,78]
Gender	Male	16 [49,50,51,52,61,62,64,67,68,70,71,72,75,76,77,78]
	Female	6 [63,65,66,73,74,79]
Interventions	Mixed Strength and conditioning program	7 [49,50,51,52,76,78,80]
	Plyometric training program	12 [61,62,63,66,67,68,70,71,73,74,75,77]
	HIIT B program	1 [79]
	Repeated sprint training	3 [64,65,70]
Outcomes	Neuromuscular power vertical (NPV)	30 [49,51,52,62,63,66,68,70,71,72,73,74,75,76,78]
	Neuromuscular power horizontal (NPH)	24 [49,50,51,61,62,66,67,68,70,71,74,77,78]
	Endurance (E)	3 [64,65,79]

**Table 4 ijerph-19-09642-t004:** Meta-regression statistical analysis for neuromuscular power vertical (NPV) according to sub-groups. In bold, models that did not reach the minimum 10 studies for meta-regression analysis. * *p* < 0.05. N = number of included studies; F = effect; Sig = Significance.

NPV		N	R Squared	Adj R Squared	Sum of Squares	F	Sig
**Training type**							
	Plyometric training	16	0.40	0.36	235.82	10.04	0.006 *
	Mixed strength and conditioning	**9**	0.73	0.69	8.31	21.23	0.002 *
**Gender**							
	Male	24	0.37	0.34	208.41	13.28	0.001 *
	Female	**5**	0.97	0.96	171.94	84.95	0.003 *
**Competitive level**							
	Elite	19	0.60	0.57	226.03	26.59	0.001 *
	Amateur	**9**	0.63	0.58	205.12	13.66	0.006 *
**Age**							
	U-16	13	0.8	0.78	144.06	47.87	0.001 *
	U-18	**8**	0.59	0.51	159.77	9.21	0.019 *

**Table 5 ijerph-19-09642-t005:** Meta-regression statistical analysis for neuromuscular power horizontal (NPH) according to sub groups. In bold, models that did not reach the minimum 10 studies for meta-regression analysis. * *p* < 0.05. N = number of included studies; F = effect; Sig = Significance.

NPH		N	R Squared	Adj R Squared	Sum of Squares	F	Sig
**Training type**							
	Plyometric training	10	0.86	0.84	588.97	53,44	0.001 *
	Mixed strength and conditioning	**6**	0.85	0.82	9.63	29.01	0.003 *
	Endurance	**5**	0.96	0.95	65.37	98.26	0.001 *
**Gender**							
	Male	19	0.83	0.83	663.03	92.40	0.001 *
	Female	**3**	0.88	0.82	35.43	14.83	0.061
**Competitive level**							
	Elite	16	0.84	0.82	479.59	76.65	0.001 *
	Amateur	**6**	0.85	0.82	164.45	29.02	0.003 *
**Age**							
	U-16	**8**	0.96	0.96	68.03	178.36	0.001 *
	U-18	**7**	0.96	0.95	221.66	141.60	0.001 *

## Data Availability

The data presented in this study are available within the article.

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
