# Peer review of "Multidisciplinary Neuromuscular and Endurance Interventions on Youth Basketball Players: A Systematic Review with Meta-Analysis and Meta-Regression"

_ijerph, 2022, doi:10.3390/ijerph19159642_

Round 1

Reviewer 1 Report

The manuscript of Arnau Sacot et al., entitled „Multidisciplinary neuromuscular and endurance interventions on youth basketball players: A systematic review with metaanalysis and meta-regression” can be interesting for the readers. However, the quality of presentation of the data must be significantly improved.

1.       Generally, the quality of Figures is very poor (I do not know whether it is due to conversion of docx to pdf, but after acceptance the article will be available in pdf format, and better quality of images is expected). Most of them seem to be „blurred”, Fig, 4. and 6 are the worst. This issue has to be solved somehow. I would strongly recommend to use higher resolution of images and in case of Fig. 4, 6, and maybe 8 as well it would be much better to structure the 4 panels simply below each other (because in the present format it decreases the possibe size of images to 50%). In case of a manuscript with 37 pages, it does not make sense to „spare” space with the 2-column structure of panels. Figure 8 it would be fine in the actual structure, if the quality would be better.

2.       Additionally, Figure legends should be in text format, and not part of the image (all of them should be changed, except Fig. 4 and 6 legends, because they are already in text format).

3.       Generally, results should be structured better, e.g. „Main search” should be point 3.1, etc. It is not clear, how „Neuromuscular interventions in youth Basketball players” (the subchapter (?) below Fig 2.) into this structure fits (should it be 3.2 or 3.1.1?). Additionally, it should be formatted better, it is not good, when it starts in the middle of the page, next to the image (and it should be probably bold and/or italic, but the structure of subchapters like 3.1., 3.1.1, 3.1.2, 3.2, 3.3 etc. could also help a lot). So, „Risk of bias assesment” could be 3.2 or 3.3, etc. Subchapters (age, gender, etc.) belonging below NPV, NPH, etc.
should be also clearly indicated as separated points (if NPV is 3.5 or similar and then the 3 age groups could be 3.5.1, 3.5.2, etc., similarly in the case of NPH, etc.). Later, meta-regression should be also clearly inserted into this structure, now it is hard to follow, where it belongs to.

4.       Generally, spell check is also needed, e.g. índex below Table 1 should be index, „inter and intra muscular” or „type II muscle fibers are grater in youth male” in discussion seems to be also reconsidered, etc.

5.       It should be indicated in the abstract as well, that youth basketball players are included in the study.

6.       It should be defined in the abstract, what „multidisciplanry interventions” mean (like it is listed in the discussion).

7.       Pylometrics / pylometric training should be also defined.

8.       Table 1 must be completely reorganized.

a.        The biggest problem is, that many of the abbreviations are simply not explained or the same abbreviation can mean more things. I would recommend e.g. M = males, and m = minutes, actually both are indicated as M. It might be supposed, that it is clear for the reader, but it is not sure. Probably everybody knows, that %=percentage, however, that is indicated below the table, and it is correct and precise in this way. But many others are not indicated, e.g. what does S5M, S10M, S20M mean? S simply is indicated as „seconds” (I would recommend to abbreviate it as „s” instead of „S”, cm instead of Cm, etc.), M could mean male or minutes, but it is clear, that in this case the abbreviation mean something completly else. If one would read all of the cited documents, it could be clear, but the aim of the table is to summarize the results of the cited articels, and it should be informative also without reading the cited documents.

b.       The format also seems to be a little bit chaotic. Somewhere it is indicated as F= female soetimes as M: male. All should be either : or =, but everywhere the same. Sometimes the members of the list are separated with comma („LJ= long jump, HJ=horizontal jump”), sometimes with ; („M: male; NPV=neuromuscular power vertical”), sometimes with nothing („IAT=Illinois agility test MPBIL= maximal power bilateral”), sometimes there is a space before or after =, sometimes not). It should be the same format everywhere.

c.        So, I would recommend to systemically go through the table, list ALL of the abbreviations below, e.g. COD and RCODA are listed, but RCOD is not, etc. CTG, CG, EG etc. are also not listed. And countless others. In this form Table 1 simply can not be understood.

d.       TCMJstep is indicated, but after = there is no explanation. Such kind of issues can be avoided, if the coauthors read the manuscript carefully before submission, it will be easier for the reader as well.

e.        It would also help a lot, to list the abbreviations in alphabetical order below the table, because there are so many of them, that otherwise it would be very hard to find, what one is looking for.

f.        Additionally, Table 1 should be cited in the text as well, where it is relevant.

9.       Table 2: It should be indicated in the legends, what one can see in the table. What do the numbers mean? Number of participants? Citations? Something else?

10.    Table 3 and 4: Similarly, like in case of % in table 1, meaning of „Sig”, N, etc. should be defined. It seems to be plausible, that the reader knows it, but not sure.

11.    Minor: „Figure 3, illustrates the forest plot for…” I think, comma is not necessary. In legends of Fig. 3., there seems to be an unnecessary space between o and f.

12.    Adolescence should be also defined, because there are differnet interpretations in the literature.

13.    „There is evidence showing that female athletes have a lower percentage of type I fibers than males and also the type II muscle fibers are grater in youth male athletes.”. This sentence seems to be contradictory. If the percentage of type I is lower in female, percentage of type II should be higher. So which parameter of type II is greater (not grater)? Diameter, mass, something else?

14.    Page numbers have to be checked as well (instead of page 7 page 2 is indicated again, on page 17 it restarts again from the 1st, and on page 21 again).

Reviewer 2 Report

Review

Multidisciplinary neuromuscular and endurance interventions on youth basketball players: A systematic review with meta-analysis and meta-regression

I welcome studies that introduce novelty and applicability the important role of neuromuscular and endurance intervention on youth basketball players. In fact, I am open to be persuaded to deep understand this relationship.

Hence, I am some sympathy with the author's intentions. In addition, the authors provide a decent description of the systematic review process, and some proposals for future research. The topic represents contemporary interest, and the scope of the work is appropriate for International Journal of Environmental Research and Public Health. I think it is very important to conduct studies like this one, because on many occasions the neuromuscular and endurance interventions, in all sports, is not taking into account when analysing the performance of youth basketball players. Therefore, it is necessary to bring to light studies whose aim is to review what has been published previously on this topic in a rigorous way. This study is particularly important because, as the authors indicate, the publications on this topic are increasingly, so it is necessary to group all these papers to have a better understanding of the results obtained previously.

From my point of view, the main strength of the manuscript is the methodology. The authors have followed properly the PRISMA guidelines. It would be interesting to check this register, but the authors do not show the link with the purpose to save the anonymity. Moreover, they have consulted the most relevant databases: Web of Science-All Databases, Medline and Cochrane databases, and the diagram flow shows the selection process.

Apart from this general commentary about the manuscript, more details of some parts of the manuscript (strengths, weakness and questions) are found hereafter.

Introduction

It is well written and structured. It is a good starting point to place the reader. However, truly think that is very short and need more information and important literature. it would be interesting if the bibliography was updated to the last five years and highlight the importance in basketball.  

Methodology

As it has been mentioned before, this section is the strongest part of the manuscript. The PRISMA methodology is clear and quality of paper. Can you show the PICOS in a table? Instead for systematic review

Discussion

In the discussion section, more detail should be provided when reporting the instruments used to assess different variables. In all case, the discussion have powerful and give a very good perspective of this research.

Conclusions

The conclusions respond to the objectives of the systematic review and meta-analysis.

Round 2

Reviewer 1 Report

The manuscript of Arnau Sacot et al., entitled „Multidisciplinary neuromuscular and endurance interventions on youth basketball players: A systematic review with metaanalysis and meta-regression” has been significantly improved with the corrections.